materials science/nanotechnology

neodymium orthoferrite, orthorhombic, perovskite, antiferromagnetic

**Author for correspondence:**
Edwin Akongnwi Nforna
e-mail: ea.nforna101@gmail.com

This article has been edited by the Royal Society of Chemistry, including the commissioning, peer review process and editorial aspects up to the point of acceptance.

# Effect of B-site Co substitution on the structure and magnetic properties of nanocrystalline neodymium orthoferrite synthesized by auto-combustion

Edwin Akongnwi Nforna[1], Patrice Kenfack Tsobnang[3], Roussin Lontio Fomekong[4], Hypolite Mathias Kamta Tedjieukeng[4], John Ngolui Lambi[4] and Julius Numbonui Ghogomu[3,2]

[1]Department of Fundamental Science, Higher Technical Teacher Training College, and
[2]Department of Chemistry, University of Bamenda, P.O. Box 39, Bambili, Cameroon
[3]Department of Chemistry, University of Dschang, P.O. Box 67, Dschang, Cameroon
[4]Department of Chemistry, Higher Teacher Training College, University of Yaounde I, P.O. Box 47, Yaounde, Cameroon

EAN, 0000-0001-8181-2585

Samples of cobalt-doped neodymium orthoferrite compounds, $NdCo_xFe_{1-x}O_3$ $(0.0 \leq x \leq 0.5)$ were synthesized via glycine auto-combustion between 250 and 300°C and calcined at 500°C for 2 h. X-ray diffraction showed that all compounds had an orthorhombic perovskite structure with space group Pbnm. Increasing cobalt doping gradually reduced the lattice parameters and contracted the unit cell volume. Both X-ray diffraction and scanning electron microscopy showed that the particles were spherical and in the nano-sized range (19–52 nm) with pores between grains. Vibrating sample magnetometry at room temperature indicated that $NdFeO_3$ has a high coercive field (1950 Oe) and cobalt substitution for iron led to a decrease in the coercive field, saturation and remanent magnetization, which was as a result of decreased magnetic moments in the crystal and reduced canting of the $FeO_6$ octahedra. The increase in magnetization and coercive fields with increase of Co was connected to the microstructure (bond lengths and angles, defects, pores, grain boundaries) and crystallite size. The compounds $NdCo_xFe_{1-x}O_3$ show antiferromagnetism with weak ferromagnetism due to

uncompensated non-collinear moments. These compounds could serve as prototypes for tuning the properties of magnetic materials (ferromagnetic and antiferromagnetic) with potential applications in data storage, logic gates, switches and sensors.

# 1. Introduction

Mixed metal oxides with the perovskite structure ($ABO_3$) have attracted much attention due to such fascinating properties as catalytic, sensor, magnetic, ferroelectric, magnetoresistive and other physical properties [1–6]. The perovskite rare-earth orthoferrites, $RFeO_3$ (R = rare-earth metal), in particular, are compounds with interesting electronic and magnetic phenomena which arise from structural features such as inter-ionic distances, types of ions present, symmetry and bond angles [2,7]. These compounds can easily be substituted in the A- and/or B-sites thus maintaining the structure while modifying the properties. The $RMO_3$ system (R = rare-earth metal, M = transition metal) is a model system for studying the magnetic interactions between the metals. Magnetic properties of these compounds are determined by three magnetic interactions M–M, M–R and R–R in descending order of strength [7–9]. The R ions show magnetic ordering only at very low temperatures, typically at T < 10 K but do not affect the basic magnetic properties of orthoferrites at high temperatures [10–12]. When M is a non-magnetic ion, the magnetic properties of the compound arise from the collective magnetic moment ordering of R at low temperatures [12]. Neodymium orthoferrite crystallizes in a distorted orthorhombic perovskite structure with four formula units per unit cell and Pbnm space group. The magnetic Fe ions are surrounded by six $O^{2-}$ ions forming a distorted octahedron. In the Fe sublattice, anisotropic magnetic ordering occurs antiferromagnetically along the a-axis. Due to the distortion of the $FeO_6$ octahedra, the antiparallel moments are not completely compensated. They are slightly canted towards the c-axis resulting in weak ferromagnetism [13]. The Neel temperature, $T_{N1}$, of $NdFeO_3$ is approximately 690 K [14]. It has been shown that at temperatures below $T_{N1}$, the Nd–Fe exchange interactions lead to the appearance of a field, $H_{Nd-Fe}$, which causes the Nd sublattice to order at very low temperatures (below 6 K) [12]. The anisotropic $H_{Nd-Fe}$ induces magnetic spin reorientations in the Fe sublattice at temperatures far below room temperature. These magnetic orderings of $NdFeO_3$ at low temperatures have been widely studied [13–17].

The most abundant magnetic oxide materials are those of the antiferromagnetic category. The problem that arises is that these antiferromagnetic materials have no commercial value, whereas ferromagnetic materials have applications in almost every electronic device such as in magnetic data storage, memory devices, switches, sensors and transformers [18]. To solve this problem, we seek to valorize these antiferromagnetic materials by tuning their structures in order to cause uncompensated antiferromagnetism also known as ferrimagnetism, thus introducing in them properties with useful applications similar to those exhibited by ferromagnetic materials. This is done in this work by the introduction of dopants and the variation of such physical parameters as temperature. Also, these materials are used for theoretical studies of magnetic phenomena.

This work, therefore, focuses on $NdFeO_3$, an antiferromagnetic material which is considered ideal for the study of different magnetic phenomena, including weak ferromagnetism. The structure and magnetic properties of this material are modified by substituting iron ions with cobalt ions. A few researchers have synthesized cobalt-doped neodymium ferrite by the ceramic [12] and sol-gel methods [2,19,20]. The properties of these compounds investigated include the spin reorientations and thermal properties at temperatures below 6 K [12,17,19], the electrical and sensor properties [2], and optical properties [20]. The magnetic properties of $NdCo_xFe_{1-x}O_3$ (x = 0, 0.03, 0.1, 0.25 and 0.5) prepared by the ceramic method showed that $Co^{3+}$ is non-magnetic at temperatures below 6 K [12]. However, cobalt ions can change from low spin to high spin and vice versa depending on the environment.

In this work, we aim to use a different synthetic method, the nitrate-glycine auto-combustion method, to prepare $NdCo_xFe_{1-x}O_3$ ($0 \leq x \leq 0.5$) and also to investigate the influence of the method of synthesis and the introduction of cobalt dopants into $NdFeO_3$ on the structure and magnetic properties. The auto-combustion method has advantages such as low temperatures and rapid synthesis and uniformity in the products obtained compared with other methods [21]. The magnetic properties, such as coercive fields, magnetization at the maximum field and remanent magnetization, are all investigated at room temperature. Since cobalt ions can change spin depending on the environment and physical properties, their spin state at room temperature is also inferred.

# 2. Material and methods

## 2.1. Preparation of undoped and cobalt-doped neodymium ferrite

The following analytical reagents were obtained from Sigma-Aldrich and used without further purification: $Nd(NO_3)_3 \cdot 6H_2O$ (99%), $Fe(NO_3)_3 \cdot 9H_2O$ (99.5%), $Co(NO_3)_2 \cdot 6H_2O$ (99%) and $C_2H_5NO_2$.

Neodymium orthoferrite and a variety of cobalt-doped neodymium orthoferrites were synthesized by the glycine auto-combustion method [21]. The appropriate amounts of the metal ions, $Nd^{3+}$, $Fe^{3+}$ and $Co^{2+}$ according to the stoichiometric mole ratios were weighed and all dissolved in 10 ml distilled water in a beaker. Stoichiometric amount of glycine was also weighed and dissolved in 5 ml distilled water in another beaker. The metals-to-glycine ratios were $3:3:10$, respectively. For example, to prepare 1.2 g of $NdFeO_3$, ($x = 0.0$, abbreviated as NFO), 2.12 g (6.42 mmol) of $Nd(NO_3)_3$ and 1.956 g (8.08 mmol) of $Fe(NO_3)_3$ were used against 1.212 g (16.14 mmol) of glycine. The cobalt-substituted neodymium orthoferrites were synthesized according to the formula $NdCo_xFe_{1-x}O_3$ with $x = 0.1$ (NCF19), $x = 0.2$ (NCF28), $x = 0.3$ (NCF37), $x = 0.4$ (NCF46) and $x = 0.5$ (NCF55). The solutions of the mixed metals and glycine were introduced into a Petri dish, stirred with a magnetic stirrer, and the mixture was gently heated while raising the temperature gradually up to the range 70–80°C until a gel was formed. Then, the temperature was further raised to the range 250–300°C, within which spontaneous ignition occurred and the reaction proceeded to the formation of the mixed metal oxides. The undoped neodymium orthoferrite formed was light brown while all the cobalt-doped samples were black, increasing in intensity with increase in cobalt content. The samples were heated at 500°C for 2 h to enhance their crystallinity. The formation of $NdFeO_3$ is given in equation (2.1) [21].

$$3Nd(NO_3)_3 + 3Fe(NO_3)_3 + 10C_2H_5NO_2 \rightarrow 3NdFeO_3 + 25H_2O + 20CO_2 + 14N_2 \qquad (2.1)$$

## 2.2. Characterization techniques

The structure and phase composition of the powdered samples were investigated by X-ray diffraction using a PANalytical X'Pert PRO 1712 diffractometer in standard Bragg–Brentano geometry with Cu $K\alpha$ ($\lambda = 1.54056$ Å) radiation. The patterns were obtained in $2\theta$ range of 10 to 80°, with a step size of 0.013. From the X-ray diffraction spectra, the crystallite size was calculated using the Scherrer formula ($D = 0.9\lambda/\beta\cos\theta$, where $\lambda$ is the source wavelength, $\beta$ is the full width at half maximum (FWHM), and $\theta$ is the reflection angle) [22]. The lattice parameters and unit cell volume were determined from the X-ray diffraction data by the Rietveld refinement method using the GSAS program [23]. The infrared (IR) spectra were obtained using the KBr method, on a SHIMADZU FTIR-84005 spectrophotometer.

In order to elucidate the morphology of the particles, a Zeiss Ultra 55 scanning electron microscope (SEM; Zeiss, Jena, Germany), was used. An energy-dispersive X-ray (EDX) spectrometer (from Oxford Instruments, Oxford, UK) was fitted with this microscope and was used for qualitative and semi-quantitative elemental analysis. For that, the experiments were carried out with a working distance of 8 mm at 15 keV. The chemical spectra were recorded with a probe current of 1 nA and the acquisition time of 300 s. The quantitative analysis of the atomic elements was done using Integrated Aztec software (AZtec 4.1 SP1, Oxford, UK).

The magnetic properties of the samples, including the coercive force $H_c$, remanent magnetization $M_r$ and saturation magnetization $M_s$, were investigated at 300 K via a vibrating sample magnetometer (VSM; model EG&G PAR 4500). Hysteresis loops were measured at room temperature (300 K) with an applied field from 0 to 15 kOe. Each sample was weighed in an aluminium foil, wrapped and introduced into the magnetometer.

# 3. Results and discussion

## 3.1. X-ray diffraction analysis

The average crystallite size, phase purity, polycrystalline structure and lattice parameters were obtained from the analysis of powder X-ray diffraction (XRD) data. The powder XRD patterns of $NdCo_xFe_{1-x}O_3$, $x = 0.0$ (NFO), 0.1 (NCF19), 0.2 (NCF28), 0.3 (NCF37), 0.4 (NCF46) and 0.5(NCF55) are shown in figure 1a.

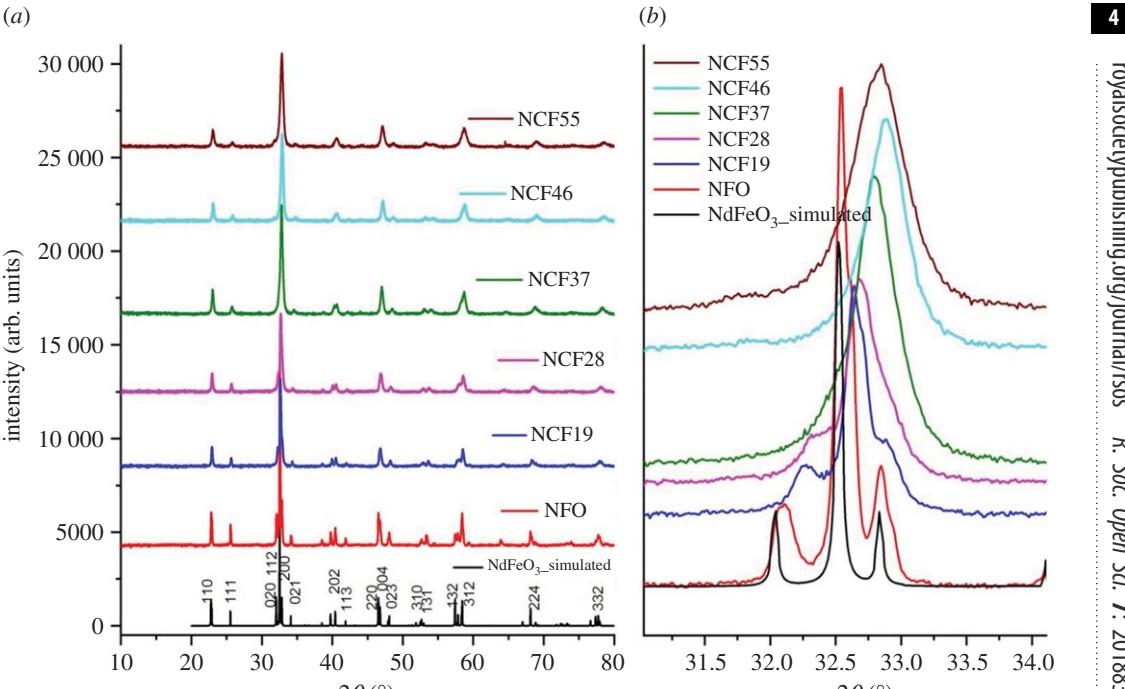

**Figure 1.** (*a*) Powder X-ray diffraction patterns of NdCo$_x$Fe$_{1-x}$O$_3$ and (*b*) zoom at the most intense peak around 32°.

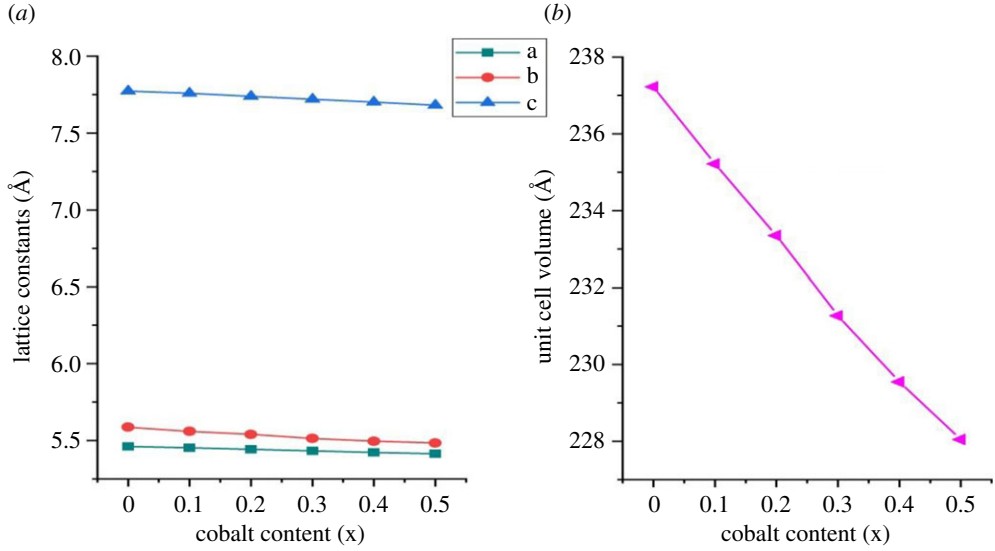

**Figure 2.** Variation of (*a*) lattice parameters and (*b*) volume with cobalt content, x.

The XRD patterns were indexed with the standard JCPDS card number 82–2421. The experimental peaks matched well with the standard with no impurity peaks thus showing single phase. These results show that the products are all perovskite oxides with orthorhombic structure and space group Pbnm (no. 62) in agreement with other researchers [2,20].

All the samples were refined in the Pbnm orthorhombic phase by the Rietveld method using the GSAS software [22,23]. The lattice parameters obtained from the Rietveld refinement are presented in table 1 and the plots of observed, calculated and difference curves in electronic supplementary material, figure S1. The lattice parameters and unit cell volume of the doped samples decreased with increase in Co content as shown in figure 2a and b.

A decrease in unit cell volume with increased Co substitution has also been reported in the literature [2,20]. This trend in lattice parameters and unit cell volume is corroborated by a shift in the XRD peaks to the right at higher angles (lower d-spacings) as the Co content is increased. The unit cell volume was

**Table 1.** Lattice and other parameters obtained from Rietveld refinement of XRD data.

| property/compound | | NdFeO$_3$ (x = 0.0, NFO) | NdCo$_{0.1}$Fe$_{0.9}$O$_3$ (x = 0.1, NCF19) | NdCo$_{0.2}$Fe$_{0.8}$O$_3$ (x = 0.2, NCF28) | NdCo$_{0.3}$Fe$_{0.7}$O$_3$ (x = 0.3, NCF37) | NdCo$_{0.4}$Fe$_{0.6}$O$_3$ (x = 0.4, NCF46) | NdCo$_{0.5}$Fe$_{0.5}$O$_3$ (x = 0.5, NCF55) |
|---|---|---|---|---|---|---|---|
| particle size (nm) | | 52.7 | 29.6 | 28.9 | 25.1 | 25.7 | 19.7 |
| lattice parameters $\alpha = \beta = \gamma = 90°$ | a (Å) | 5.4622(17) | 5.4531(4) | 5.4423(5) | 5.4333(6) | 5.4234(8) | 5.4139(11) |
| space group Pbnm centric primitive orthorhombic | b (Å) | 5.5867(19) | 5.5597(4) | 5.5406(5) | 5.5134(5) | 5.4959(7) | 5.4842(12) |
| | c (Å) | 7.7738(23) | 7.7585(6) | 7.7388(7) | 7.7204(8) | 7.7014(10) | 7.6810(16) |
| unit cell volume (Å$^3$) | | 237.224(13) | 235.218(29) | 233.353(35) | 231.27(4) | 229.55(6) | 228.05(8) |
| Fe—O1—Fe (°) | | 151.274(1) | 152.707(3) | 154.643(3) | 156.130(3) | 160.348(3) | 162.238(5) |
| Fe—O2—Fe (°) | | 151.160(1) | 152.245(1) | 154.232(2) | 155.412(2) | 157.890(2) | 162.000(3) |
| Fe/Co—O1 (Å) | | 2.00615(6) | 1.89346(13) | 1.82236(16) | 1.76616(17) | 1.70917(22) | 1.62618(32) |
| Fe/Co—O2 (Å) | | 1.96747(4) | 1.87647(10) | 1.75290(11) | 1.74405(13) | 1.72302(18) | 1.69993(25) |

**Table 2.** Goldschmidt's tolerance factor for $NdCo_xFe_{1-x}O_3$.

| compound | NFO | NCF19 | NCF28 | NCF37 | NCF46 | NCF55 |
|---|---|---|---|---|---|---|
| tolerance factor, $t$ | 0.923 | 0.928 | 0.932 | 0.937 | 0.942 | 0.946 |

expected to increase as the initial reactant, $Co^{2+}$ ions, (ionic radius, 0.74 Å) is substituting the smaller $Fe^{3+}$ ions (0.645 Å) [24]. The energy differences between different electronic configurations of d-metal ions is close; therefore, different valence states and electronic configurations are easily inter-convertible [25].

As a consequence, when $Co^{2+}$ ions substitute $Fe^{3+}$ ions in the crystal, the $Co^{2+}$ ions are oxidized to $Co^{3+}$ ions thereby maintaining electrical neutrality. The resulting decrease in unit cell volume is therefore due to the presence of smaller $Co^{3+}$ ions (low spin, LS 0.545 Å, high spin, HS 0.61 Å).

In the Rietveld refinement of $NdFeO_3$ perovskite, the unit cell consists of the $Fe^{3+}$ ions fixed at the 4(b) Wyckoff position, the $Nd^{3+}$ ions at the 4(c) position and $O^{2-}$ ions occupy the 4(c) and 4(d) positions. There are four $Fe^{3+}$ ions per unit cell. Each $Fe^{3+}$ ion is surrounded by six $O^{2-}$ ions forming an octahedron while the $Nd^{3+}$ ions fit in the interstices of the $FeO_6$ octahedra in dodecahedral coordination [26]. $Co^{3+}$ ions enter the $Fe^{3+}$ ions site in the doped samples.

The ideal $ABO_3$ perovskite crystallizes in a cubic lattice with B—O—B angles of 180°. Some distortions may exist in the ideal cubic perovskite, to result in the formation of orthorhombic, rhombohedral, hexagonal and tetragonal forms. These distortions arise from three aspects: (i) sizes of the cations, (ii) deviations from the ideal composition, and (iii) Jahn–Teller effect [25,26]. The differences in the cation sizes could be evaluated by the Goldschmidt's tolerance factor [25]. It is a real measure of the degree of the distortion of perovskite from the ideal cubic structure so the value of $t$ tends to unity as the structure adopts the cubic form.

$$t = \frac{(r_A + r_O)}{\sqrt{2}(r_B + r_O)}.$$

The tolerance factor for $NdCo_xFe_{1-x}O_3$ is presented in table 2. Generally, the mismatch between the Nd—O and Fe—O bond lengths introduces internal stresses which result in tilting of the $FeO_6$ octahedra. The Fe—O–Fe bond angle is a measure of the tilting of the octahedron [26–28]. This is seen in the Fe—O—Fe angles in all the samples being less than ideal cubic perovskite value of 180°. The Fe/Co—O bond lengths from table 1 decrease with Co content while the Fe—O—Fe angles increase. It indicates that the octahedral distortion is reduced with Co doping. This is because introducing smaller $Co^{3+}$ ions in the crystal decrease the internal stress, as illustrated by a shift in the value of the tolerance factor towards unity.

From the patterns, the average crystallite size was determined from the (110), (111), (020), (112), (200) (022), (220) and (312) peaks using the Scherrer formula. The results presented in table 1 show that all the particles are in the nano-size range. The average crystallite size decreases with increasing cobalt substitution as indicated in the literature [2]. This is due to the incorporation of the smaller $Co^{3+}$ ions (LS 0.545 Å, HS 0.61 Å) into the crystal lattice in place of the larger $Fe^{3+}$ ions (HS 0.645 Å) [24]. Figure 1b, a zoom of the most intense peak, shows that as the Co composition increases, the peak becomes broader and the triplet peak merges to a single broad peak. This confirms the fact that the particles are nano-size, as peak broadening is related to reduced particle size and/or strain. In this study, larger crystallites were obtained using the auto-combustion method and after calcination at 500°C compared with the sol-gel method after calcination at 700°C [20] and 800°C [2].

## 3.2. Infrared analysis

Qualitative analysis of the functional groups present in the samples was carried out using FTIR spectroscopy. FTIR spectra with wavenumbers in the range 600–4000 $cm^{-1}$ were obtained for the $NdCo_xFe_{1-x}O_3$ nanopowders calcined at 500°C and the spectra are presented in electronic supplementary material, figure S2. The major peaks in the frequency range of 615–660 $cm^{-1}$ are shown in table 3. These peaks correspond to the stretching vibrations of the M–O bond (M = Nd or Co) [20]. From XRD data, the bond lengths decrease with increased Co, hence higher bond strengths. This results in a shift to higher frequencies in the FTIR spectra. The stretching vibration frequencies of ferric oxide which usually occurring around 300, 400 and 500 $cm^{-1}$ are absent, because they fall below the detection limit of the instrument [29,30]. M–O bond frequencies generally occur between 270 and

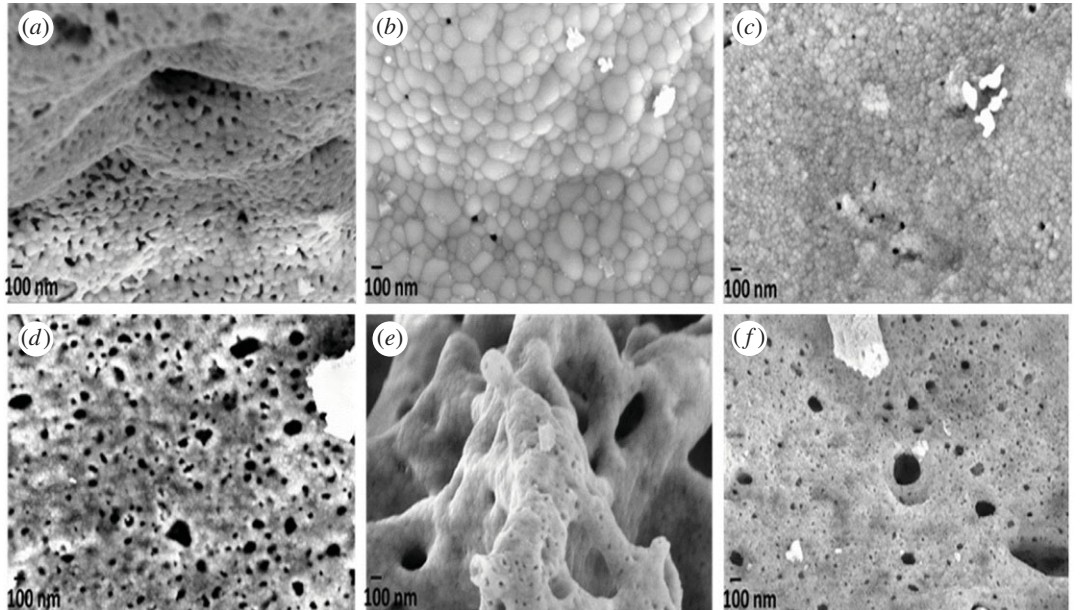

**Figure 3.** SEM images of $NdCo_xFe_{1-x}O_3$ at 100 nm: (*a*) NFO, (*b*) NCF19, (*c*) NCF28, (*d*) NCF37, (*e*) NCF46 and (*f*) NCF55.

**Table 3.** Important IR absorption frequencies for studied samples.

| compound | NFO | NCF19 | NCF28 | NCF37 | NCF46 | NCF55 |
|---|---|---|---|---|---|---|
| major band ($cm^{-1}$) | 616 | 626 | 628 | 629 | 636 | 654 |

750 $cm^{-1}$, while metal nitrates exhibit higher frequencies between 730 and 2450 $cm^{-1}$ [30,31]. No peaks were observed above 700 $cm^{-1}$, indicating that there was neither a nitrate nor OH group present. These results are, therefore, in agreement with those from XRD.

## 3.3. Scanning electron microscopy analysis

In order to investigate the morphology and microstructure of the compounds, scanning electron micrographs were taken, and the SEM images of the mixed metal ferrites after calcinations at 500°C are shown in figure 3*a–f*. The microstructural investigation reveals spherical particles. It is also observed that the particles have a tendency of agglomeration. This phenomenon in orthoferrites has been reported by several researchers [32,33]. The particles are of polycrystalline crystallites. Due to agglomeration, the average grain sizes vary in the different compounds. Agglomeration is highly advanced in the following compounds: NFO and NCF19, hence they have the largest particle sizes. It is evident that increasing cobalt doping provokes decreasing particle size which is in agreement with the XRD results.

The microstructure also reveals that the synthesis method adopted resulted in products with pores. The presence of pores in $NdFeO_3$ has been reported in the literature [34]. This morphology is sharply different from that reported for $NdCo_xFe_{1-x}O_3$ via the sol-gel method [20]. It can be clearly seen that NFO and NCF19 have large particles with few pores. The porosity is high in NCF28 to NCF55. The presence of pores in the microstructure arises from the differences in the tilted $FeO_6$ and $CoO_6$ octahedra.

## 3.4. Energy-dispersive X-ray spectroscopy

The qualitative and semi-quantitative elemental analysis for NFO, NCF19, NCF28, NCF37, NCF46 and NCF55 were carried out using EDX. The EDX spectra for all the samples are presented in figure 4*a–f*. An analysis of the data for all the samples is summarized in table 4. Several regions were analysed per sample and the results close to the stoichiometry were chosen.

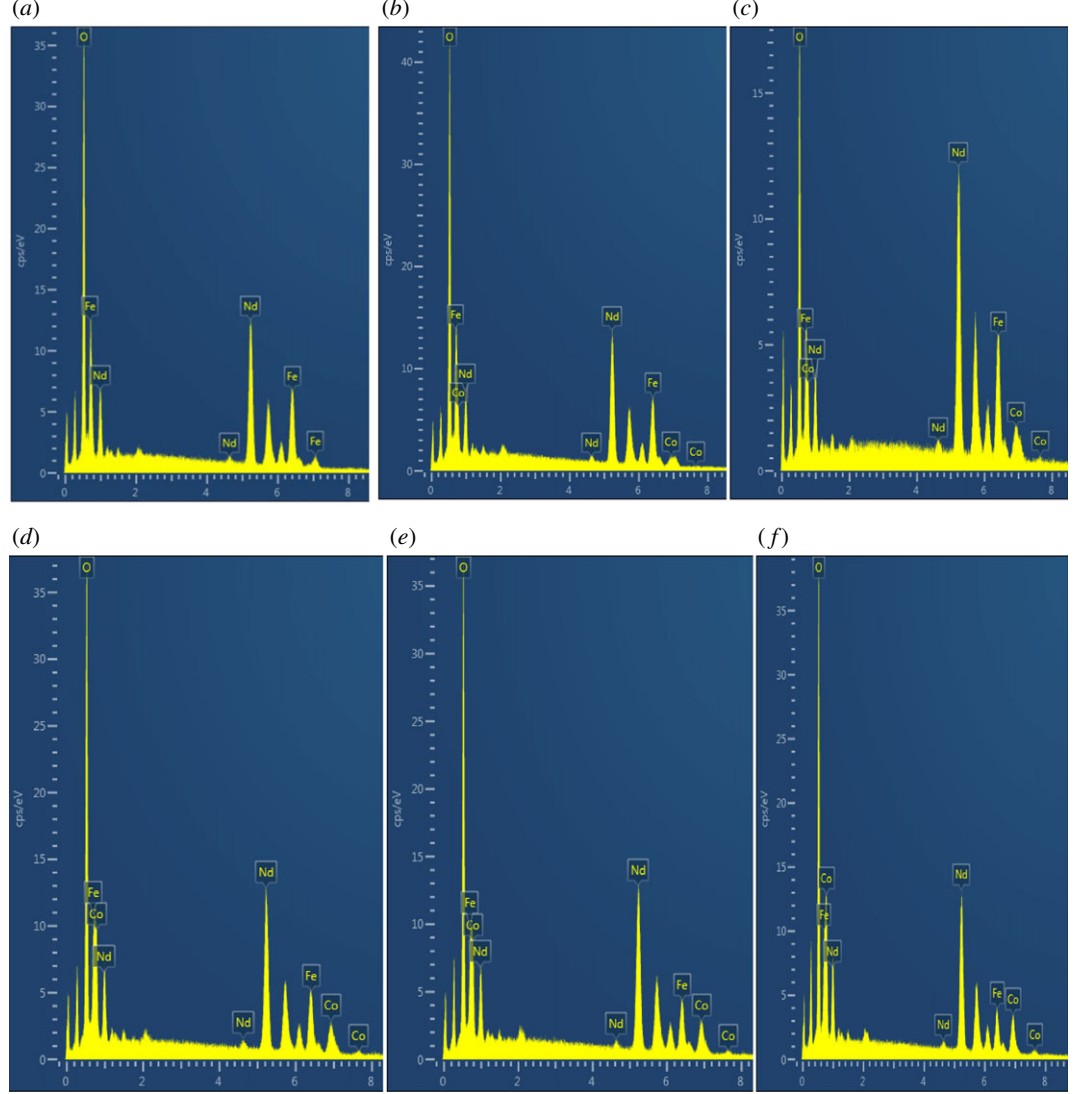

**Figure 4.** EDX spectra of (*a*) NFO, (*b*) NCF19, (*c*) NCF28, (*d*) NCF37, (*e*) NCF46 and (*f*) NCF55.

The EDX spectra were used to analyse the metal contents qualitatively and quantitatively. Qualitatively, the spectrum for NFO showed the presence of only Nd, Fe and O elements while all the doped samples also showed in addition the presence of Co. The experimental mole ratios of the metals to the nearest whole numbers match with the expected mole ratios. The experimental errors are thus small.

Moving from $x = 0.1$ to 0.5, the number of moles of cobalt increases while the number of moles of iron decreases as expected. Number of moles of Nd is constant for all the compounds. Therefore, the expected composition was attained, confirming the efficiency of the synthesis technique.

## 3.5. Magnetic properties by vibrating sample magnetometer

Magnetic hysteresis loops for magnetic measurements at room temperature by VSM with a maximum applied field of 15 kOe are presented in figure 5 for all the samples. From the figure, it is seen that all the samples exhibit spontaneous magnetization from an initial value of zero. The narrow hysteresis loops are an indication of weak ferromagnetism. It has been shown that in $NdFeO_3$, $Nd^{3+}$ magnetic moments order at very low temperatures, far below room temperature, while at room temperature, the magnetic properties of $NdFeO_3$ are determined by the ordering of $Fe^{3+}$ magnetic moments [13,14,19]. Rare-earth orthoferrites, $RFeO_3$ in general (e.g. $NdFeO_3$), are reported to exhibit non-collinear antiferromagnetism at room temperature. Below the Neel temperature, $Fe^{3+}$ sublattice shows a $\{G_x, M_z\}$-type magnetic ordering leading to a canted antiferromagnetic structure with a small total

**Table 4.** EDX atomic % and mole ratios for $NdCo_xFe_{1-x}O_3$. Ex. = experimental; Exp. = expected.

| sample | atomic % | | | | metal mole ratios | | | | | | | | |
|---|---|---|---|---|---|---|---|---|---|---|---|---|---|
| | Nd | Co | Fe | O | Nd | | | Co | | | Fe | | |
| | Ex. | Ex. | Ex. | Ex. | Ex. | Exp. | Error | Ex. | Exp. | Error | Ex. | Exp. | Error |
| NF0 | 21.6 | 0 | 20.42 | 57.98 | 1.08 | 1.0 | 0.08 | — | — | — | 1.02 | 1.0 | 0.02 |
| NCF19 | 20.44 | 1.77 | 17.90 | 59.90 | 1.02 | 1.0 | 0.02 | 0.09 | 0.1 | 0.01 | 0.90 | 0.9 | 0.0 |
| NCF28 | 22.69 | 4.86 | 17.32 | 55.13 | 1.13 | 1.0 | 0.13 | 0.24 | 0.2 | 0.02 | 0.87 | 0.8 | 0.07 |
| NCF37 | 20.84 | 7.41 | 13.99 | 57.77 | 1.04 | 1.0 | 0.04 | 0.37 | 0.3 | 0.07 | 0.70 | 0.7 | 0.0 |
| NCF46 | 21.90 | 7.89 | 12.03 | 58.18 | 1.09 | 1.0 | 0.09 | 0.39 | 0.4 | 0.01 | 0.60 | 0.6 | 0.0 |
| NCF55 | 21.21 | 9.90 | 9.76 | 59.13 | 1.06 | 1.0 | 0.06 | 0.49 | 0.5 | 0.01 | 0.49 | 0.5 | 0.01 |

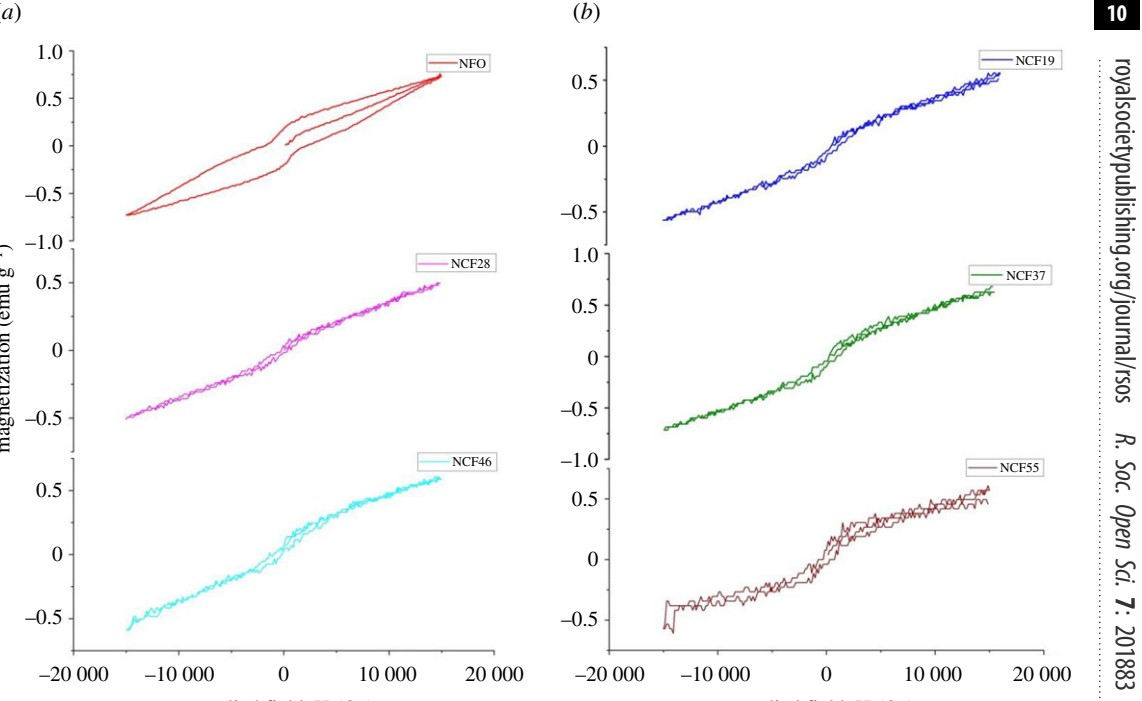

**Figure 5.** Hysteresis loops of $NdCo_xFe_{1-x}O_3$ at room temperature.

ferromagnetic moment **F** directed along the **c** crystal axis (parallel to the z-axis) and an antiferromagnetic vector **G** directed along the **a** crystal axis (parallel to the a-axis) [3,13].

An understanding of magnetism in $NdFeO_3$ (NFO) is connected to the localized unpaired d electrons and the microstructure. $Fe^{3+}$ is a $3d^5$ metal ion with five unpaired electrons ($t_{2g}^3 e_g^2$ $S = 5/2$) at the high spin (HS) state. In the Fe sublattice, long-range magnetic ordering occurs where spin–spin coupling involves antiferromagnetic coupling between nearest neighbour $Fe^{3+}$ ions by the exchange of electrons with the intervening $O^{2-}$ ion, Fe–O–Fe interactions known as superexchange interaction [25,35]. As confirmed by XRD, $FeO_6$ octahedra are tilted, which results in non-collinearity of the antiparallel spins. Therefore, a canted weak ferromagnetism is found in $NdFeO_3$ as observed for similar ferrites in the literature [13,36,37].

The doping of $NdFeO_3$ with cobalt not only influences the structural features but also has an effect on the saturation magnetization, $M_s$, coercive field, $H_c$, and remanent magnetization, $M_R$, as shown in figure 6a and b.

The saturation magnetization, $M_s$ for all the compositions was obtained from the plot of M versus 1/H, for high H values, and extrapolating to 1/H = 0 (electronic supplementary material, figure S3) [38,39]

It is observed that Co-substitution in $NdFeO_3$ greatly decreases the magnetization, coercive field, $H_c$ and remanent magnetization, $M_R$. From the graph, the saturation magnetization of $NdFeO_3$ is 1.43 emu g$^{-1}$. $Co^{3+}$ is a $3d^6$ metal ion, which, depending of its environmental conditions such as temperature, pressure and ionic crystal, could exhibit three spin states: low spin, LS ($t_{2g}^6 e_g^0$ $S = 0$), intermediate spin, IS ($t_{2g}^5 e_g^1$ $S = 1$) and high spin, HS ($t_{2g}^4 e_g^2$ $S = 2$) [40]. The saturation magnetization is known to depend on the number of magnetic atoms per unit volume and the magnitude of atomic magnetic moments [41]. Therefore, the lower magnetization of the cobalt-doped $NdFeO_3$ compounds compared with the undoped is due to the replacement of iron ions of higher magnetic moments with cobalt ions of lower magnetic moments. Also, the decrease of the tilted angle of the canted $FeO_6$ octahedra upon substitution of Fe with Co favours antiferromagnetism and therefore decreases the overall magnetization.

The saturation magnetization, $M_s$, generally increases from $x = 0.1$, NCF19 (0.71 emu g$^{-1}$) to 0.5, NCF55 (1.28 emu g$^{-1}$), with the exception of NCF28 (0.87 emu g$^{-1}$) which is very high. The remanent magnetization, $M_R$ also increases with cobalt content from 0.03 Oe (NCF19) to 0.06 Oe (NCF55). The magnetic interactions in the crystal become complex with the introduction of $Co^{3+}$ ions. The presence of cobalt ions in $NdCo_xFe_{1-x}O_3$ as $Co^{3+}$ in low spin state at very low temperatures has been reported in the literature [12,19]. One would expect the magnetization to decrease with increasing $Co^{3+}$ if it is present in the low spin state. Also, the decrease in distortion as confirmed by the powder XRD results should favour antiferromagnetism and therefore decrease the magnetization with increasing Co content.

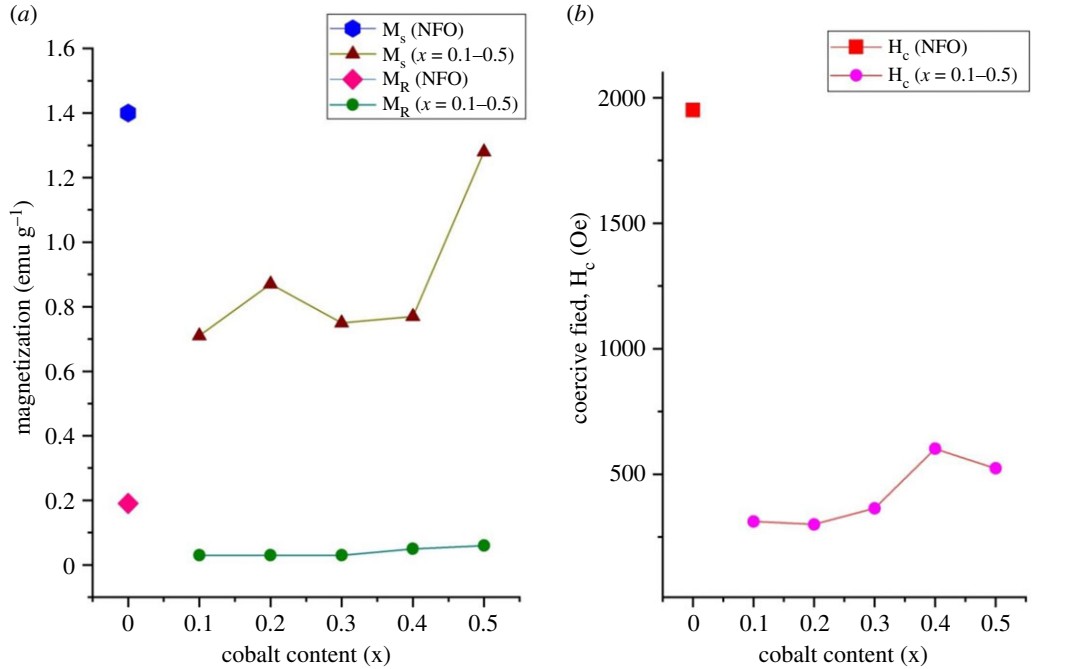

**Figure 6.** (a) Variation of $M_s$ and $M_R$ of $NdCo_xFe_{1-x}O_3$ samples with x and (b) variation of coercive field of $NdCo_xFe_{1-x}O_3$ samples with x.

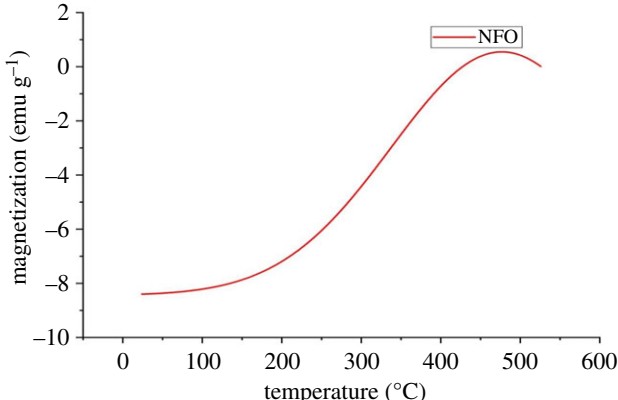

**Figure 7.** Temperature variation of magnetization measured in a field of 200 Oe.

The observed trend in $M_s$ could be attributed to large grain boundaries with a decrease in grain size and magnetic inclusions [41] of the cobalt phase that cause uncompensation of the antiparallel spins. The sudden increase in $M_s$ of NCF28 could be attributed to defects in the nanocrystallites (the micrograph of figure 2c), which result in large uncompensated antiparallel spins and anisotropy strain at the surface favouring ferromagnetism [41,42].

The coercive field of $NdFeO_3$ is 1950 Oe which is higher than reported in the literature [43]. There is a general increase in the coercivity from NCF19 (312 Oe) to NCF55 (523 Oe) with anomalies at $x = 0.2$, NCF28 (slightly lower than NCF37) and $x = 0.4$, NCF46 (higher than NCF55).

Coercivity is related to the microstructure and magnetic anisotropy energy. The coercive field increases when it is difficult to move domain walls. Domain wall motion is restricted to structural defects such as inclusions, voids and precipitates of a non-magnetic phase [38,41,42]. The general increase in the coercive fields with cobalt content coincides with a decrease in particle size (with more pinned domains walls) and presence of pores, inclusions in the microstructure. This implies that the magnetic properties of the Co-doped $NdFeO_3$ are more stable at room temperature with an increase in the amount of Co. The anomalies may be due to defects and varied magnetocrystalline anisotropy in the crystallites.

The temperature dependence on magnetization measured from room temperature (23°C) to 520°C is given in figure 7. The temperature dependence of magnetization recorded while warming in a low field

(200 Oe) of freshly prepared samples is considered a zero-field cooled (ZFC) magnetization and shows different characteristics [44]. The shape and nature of the curve could be used to determine the Neel temperature and magnetic anisotropy. The magnitude and temperature dependence of the coercive field of a compound is reflected in the shape of the corresponding ZFC magnetization curve [44]. The figure shows a broad change in magnetization with a maximum. This indicates large magnetocrystalline anisotropy or a high resistance to domain wall motion and hence high coercive field for NFO. From the graph, it is seen that the transition into the paramagnetic phase (Neel temperature) falls in the range given in the literature [14,37].

# 4. Conclusion

Undoped and cobalt-doped neodymium ferrites were prepared by glycine auto-combustion. All the samples had orthorhombic perovskite structures with spherical nanoparticles. The cobalt doping resulted in a decrease in the lattice parameters and the unit cell volume. There was also a decrease in magnetization when cobalt was introduced into the crystal lattice and increasing cobalt content resulted in coercive fields increased in the doped samples. The magnetic properties show that these compounds show uncompensated antiferromagnetism with an overall weak ferromagnetism that arises from the microstructure and crystallite sizes. Thus, doping neodymium orthoferrite with cobalt is, in general, a suitable way of tuning the perovskite structure. It is also a way of modifying, in particular, magnetic properties by introducing magnetic inclusions which may render them suitable for applications in data storage, logic gates, switches and sensors.

Ethics. This article does not present research with ethical considerations

Data accessibility. Additional information concerning this paper is available in the electronic supplementary material and in Dryad Digital Repository. https://doi.org/10.5061/dryad.g79cnp5nw [45].

Authors' contributions. E.A.N. designed the study, prepared the nanopowder samples, performed XRD and VSM, and wrote the draft manuscript. P.K.T. analysed FTIR and XRD spectra; R.L.F. carried out SEM and EDX; H.M.K.T., J.N.L. and J.N.G. edited the manuscript. All authors gave final approval for publication and agree to be held accountable for the work performed therein.

Competing interests. The authors declare no competing interest.

Funding. Funding was from authors' individual incomes.

Acknowledgements. Part of this work was carried out in the Magnetic Materials Unit, at the Physical and Materials Chemistry Division, CSIR-National Chemical Laboratory, Pune, India. Our sincere gratitude is hereby addressed to Dr P. A. Joy, Head of this Research Unit.

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
