## [Peer Review File · Royal Society Open Science]

Review History

RSOS-201883.R0 (Original submission)

Review form: Reviewer 1

Is the manuscript scientifically sound in its present form?

No

Are the interpretations and conclusions justified by the results?

No

Is the language acceptable?

Yes

Do you have any ethical concerns with this paper?

No

Have you any concerns about statistical analyses in this paper?

No

Recommendation?

Major revision is needed (please make suggestions in comments)

Comments to the Author(s)

Title: Effect of B-site Co substitution on the structure and magnetic properties of nanocrystalline neodymium orthoferrite synthesized by auto-combustion

Manuscript ID: RSOS-201883

In this article, cobalt-doped neodymium orthoferrite compounds, $\text{NdCo}_x\text{Fe}_{1-x}\text{O}_3$ ($0.0 \leq x \leq 0.5$) were synthesized via glycine auto-combustion between 250 – 300 °C and calcined at 500 °C for two hours. X-ray diffraction showed that $\text{NdCo}_x\text{Fe}_{1-x}\text{O}_3$ ($0.0 \leq x \leq 0.5$) samples had an orthorhombic perovskite structure with space group $Pbnm$. It was noted that increasing cobalt doping gradually reduced the lattice parameters and contracted the unit cell volume. The paper should be revised before the publication.

- The sample coding should be included in the manuscript.
 - Instead of FTIR Table, FTIR Spectrum may be included
 - Change Morphology in SEM with Cobalt content should be explained in detail
 - EDX spectrum is need to all the prepared samples and possible to correlate the stoichiometric ratio estimated from EDX graph.
 - It not possible to accept the VSM and Magnetization versus temperature graph
 - It is difficult to measure the Saturation Magnetisation by this VSM. Experiment should be performed with high magnetic field like 75 Koe. Temperature dependent Magnetization should be done for all the samples with and without Field cooling
- Therefore, the paper should be revised majorly with the proper results and recent reference before publication.

Review form: Reviewer 2

Is the manuscript scientifically sound in its present form?

No

Are the interpretations and conclusions justified by the results?

No

Is the language acceptable?

Yes

Do you have any ethical concerns with this paper?

No

Have you any concerns about statistical analyses in this paper?

No

Recommendation?

Major revision is needed (please make suggestions in comments)

Comments to the Author(s)

The manuscript submitted by Nforna et al. report the “Effect of B-site Co substitution on the structure and magnetic properties of nanocrystalline neodymium orthoferrite synthesized by auto-combustion.” In this report, authors have synthesized cobalt-doped neodymium orthoferrite

compounds, $\text{NdCo}_x\text{Fe}_{1-x}\text{O}_3$ ($0.0 \leq x \leq 0.5$) using auto-combustion method which is different from the method have been reported previously to prepare $\text{NdCo}_x\text{Fe}_{1-x}\text{O}_3$. The powder XRD, EDX and magnetic measurements were performed which shows difference in structural and magnetic behavior of the reported compounds in comparison to the undoped NdFeO . However, effect of doped Co amount on the both (structural and magnetic) follows an ambiguous trend and not well understood. Additionally, similar type of works has been previously reported. The manuscript needs several changes before publication in the high standard Royal Society Open Science Journal.

No IR Figure have shown in main text and SI, author should put IR figure of all reported samples in SI.

In Figure 1 caption, author should write the formula correctly.

It is obvious that the difference in stretching vibration of $\text{M}-\text{O}$ in NCF19 and NCF55 is about 30 cm^{-1} . Can the authors explain the reason?

The $\text{Fe}-\text{O}-\text{Fe}$ bond angles for all reported compounds lie in the range of 150 -160° (except NCF46) indicating significant deviation in FeO_6 octahedral geometry not slightly. Author should modify the sentence accordingly.

Kindly change the format of Figure 3, 3a and 3b for NOF and NCF55 according to the text which is just reverse in Figure 3.

It is obvious from the PXRD data and magnet data that there are several mismatches in the reported trend in structural and magnetic properties (e.g. particle size, $\text{M}-\text{O}$ distances, $\text{Fe}-\text{O}-\text{Fe}$ angles, M_S , M_R , H_c , etc.) upon Co doping. Can the authors explain the reasons?

Why the Hysteresis loops of $\text{NdCo}_x\text{Fe}_{1-x}\text{O}_3$ at room temperature (Figure 4) are so noisy?

Figure 5. Magnetization versus temperature graph of NFO to determine the Neel temperature, they need to remeasure this data. This data is not acceptable in this current form. From this data it is impossible to find out the Neel temperature at 421 °C (694 K).

Co^{3+} in LS state is diamagnetic and have no contribution to the magnetic properties. Then how it affects the exchange interactions?

There are several errors in reference number, Table number, and Figure number formats.

In general, many spaces are missing in between the words in the text. Kindly check this.

Decision letter (RSOS-201883.R0)

Dear Dr Nforna:

Title: Effect of B-site Co substitution on the structure and magnetic properties of nanocrystalline neodymium orthoferrite synthesized by auto-combustion
 Manuscript ID: RSOS-201883

The editor assigned to your manuscript has now received comments from reviewers. We would like you to revise your paper in accordance with the referee and Subject Editor suggestions which can be found below (not including confidential reports to the Editor). Please note this decision does not guarantee eventual acceptance.

Please submit your revised paper before 13-Dec-2020. Please note that the revision deadline will expire at 00.00am on this date. If we do not hear from you within this time then it will be assumed that the paper has been withdrawn. In exceptional circumstances, extensions may be possible if agreed with the Editorial Office in advance. We do not allow multiple rounds of revision so we urge you to make every effort to fully address all of the comments at this stage. If deemed necessary by the Editors, your manuscript will be sent back to one or more of the original reviewers for assessment. If the original reviewers are not available we may invite new reviewers.

On behalf of the Subject Editor Professor Anthony Stace and the Associate Editor Dr Dattatray Late.

RSC Associate Editor:
Comments to the Author:
Major Revision needed

RSC Subject Editor:
Comments to the Author:

(There are no comments.)

Reviewers' Comments to Author:

Reviewer: 1

Comments to the Author(s)

Title: Effect of B-site Co substitution on the structure and magnetic properties of nanocrystalline neodymium orthoferrite synthesized by auto-combustion

Manuscript ID: RSOS-201883

In this article, cobalt-doped neodymium orthoferrite compounds, $\text{NdCo}_x\text{Fe}_{1-x}\text{O}_3$ ($0.0 \leq x \leq 0.5$) were synthesized via glycine auto-combustion between 250 – 300 °C and calcined at 500 °C for two hours. X-ray diffraction showed that $\text{NdCo}_x\text{Fe}_{1-x}\text{O}_3$ ($0.0 \leq x \leq 0.5$) samples had an orthorhombic perovskite structure with space group Pbnm. It was noted that increasing cobalt doping gradually reduced the lattice parameters and contracted the unit cell volume. The paper should be revised before the publication.

- The sample coding should be included in the manuscript.
 - Instead of FTIR Table, FTIR Spectrum may be included
 - Change Morphology in SEM with Cobalt content should be explained in detail
 - EDX spectrum is need to all the prepared samples and possible to correlate the stoichiometric ratio estimated from EDX graph.
 - It not possible to accept the VSM and Magnetization versus temperature graph
 - It is difficult to measure the Saturation Magnetisation by this VSM. Experiment should be performed with high magnetic field like 75 Koe. Temperature dependent Magnetization should be done for all the samples with and without Field cooling
- Therefore, the paper should be revised majorly with the proper results and recent reference before publication.

Reviewer: 2

Comments to the Author(s)

The manuscript submitted by Nforna et al. report the “Effect of B-site Co substitution on the structure and magnetic properties of nanocrystalline neodymium orthoferrite synthesized by auto-combustion.” In this report, authors have synthesized cobalt-doped neodymium orthoferrite compounds, $\text{NdCo}_x\text{Fe}_{1-x}\text{O}_3$ ($0.0 \leq x \leq 0.5$) using auto-combustion method which is different from the method have been reported previously to prepare $\text{NdCo}_x\text{Fe}_{1-x}\text{O}_3$. The powder XRD, EDX and magnetic measurements were performed which shows difference in structural and magnetic behavior of the reported compounds in comparison to the undoped NdFeO . However, effect of doped Co amount on the both (structural and magnetic) follows an ambiguous trend and not well understood. Additionally, similar type of works has been previously reported. The manuscript needs several changes before publication in the high standard Royal Society Open Science Journal.

No IR Figure have shown in main text and SI, author should put IR figure of all reported samples in SI.

In Figure 1 caption, author should write the formula correctly.

It is obvious that the difference in stretching vibration of M–O in NCF19 and NCF55 is about 30 cm^{-1} . Can the authors explain the reason?

The Fe—O—Fe bond angles for all reported compounds lie in the range of 150 -160° (except NCF46) indicating significant deviation in FeO₆ octahedral geometry not slightly. Author should modify the sentence accordingly.

Kindly change the format of Figure 3, 3a and 3b for NOF and NCF55 according to the text which is just reverse in Figure 3.

It is obvious from the PXRD data and magnet data that there are several mismatches in the reported trend in structural and magnetic properties (e.g. particle size, M—O distances, Fe—O—Fe angles, MS, MR, Hc, etc.) upon Co doping. Can the authors explain the reasons?

Why the Hysteresis loops of NdCo_xFe_{1-x}O₃ at room temperature (Figure 4) are so noisy?

Figure 5. Magnetization versus temperature graph of NFO to determine the Neel temperature, they need to remeasure this data. This data is not acceptable in this current form. From this data it is impossible to find out the Neel temperature at 421 °C (694 K).

Co³⁺ in LS state is diamagnetic and have no contribution to the magnetic properties. Then how it affects the exchange interactions?

There are several errors in reference number, Table number, and Figure number formats.

In general, many spaces are missing in between the words in the text. Kindly check this.

Author's Response to Decision Letter for (RSOS-201883.R0)

See Appendix A.

Decision letter (RSOS-201883.R1)

This year has been very difficult for everyone, and we want to take the opportunity to thank you for your continued support in 2020.

The Royal Society Open Science editorial office will be closed from the evening of Friday 18 December 2020 until Monday 4 January 2021. We will not be responding during this time. If you have received a deadline within this time period, please contact us as soon as possible to allow us to extend the deadline. If you receive any automated messages during this time asking you to meet a deadline, we offer apologies and invite you to respond after the festive period or during normal working hours.

With our best for a peaceful festive period and New Year, and we look forward to working with you in 2021.

Dear Dr Nforna:

Title: Effect of B-site Co substitution on the structure and magnetic properties of nanocrystalline neodymium orthoferrite synthesized by auto-combustion

Manuscript ID: RSOS-201883.R1

It is a pleasure to accept your manuscript in its current form for publication in Royal Society Open Science. The chemistry content of Royal Society Open Science is published in collaboration with the Royal Society of Chemistry.

On behalf of the Subject Editor Professor Anthony Stace and the Associate Editor Dr Dattatray Late.

RSC Associate Editor
Comments to the Author:
Authors have revised the manuscript as per comments and now suitable for publication.

Reviewer(s)' Comments to Author:

Appendix A

The Editorial Board
Royal Society Open Science

Dear Editor

Authors Response to Referees on Manuscript RSOS-201883, title: “Effect of B-site Co substitution on the structure and magnetic properties of nanocrystalline neodymium orthoferrite synthesized by auto-combustion”

We extend our gratitude to the editorial board and the reviewers for the comments and suggestions which we have meticulously considered for improving the manuscript.

RSC Associate Editor:

Comments to the Author:

Major Revision needed

We thank the Associate Editor for the assessment of the manuscript. We have revised the manuscript following the reviewers’ comments.

Reviewers' Comments to Author:

Reviewer: 1

Comments to the Author(s)

Title: Effect of B-site Co substitution on the structure and magnetic properties of nanocrystalline neodymium orthoferrite synthesized by auto-combustion

Manuscript ID: RSOS-201883

In this article, cobalt-doped neodymium orthoferrite compounds, $\text{NdCo}_x\text{Fe}_{1-x}\text{O}_3$ ($0.0 \leq x \leq 0.5$) were synthesized via glycine auto-combustion between 250 – 300 °C and calcined at 500 °C for two hours. X-ray diffraction showed that $\text{NdCo}_x\text{Fe}_{1-x}\text{O}_3$ ($0.0 \leq x \leq 0.5$) samples had an orthorhombic perovskite structure with space group Pbnm. It was noted that increasing cobalt doping gradually reduced the lattice parameters and contracted the unit cell volume. The paper should be revised before the publication.

→ The sample coding should be included in the manuscript.

We thank the reviewer for the suggestion. We have included the codes for all the compounds in sections 2.1 and 3.1.

→ Instead of FTIR Table, FTIR Spectrum may be included

We have included the FTIR spectra for all the samples in the supplementary information, SI (figure S2) and referenced in the text.

→ **Change Morphology in SEM with Cobalt content should be explained in detail**

We thank the reviewer for the suggestion. We have explained in details how the morphology changes with the change in cobalt content.

→ **EDX spectrum is need to all the prepared samples and possible to correlate the stoichiometric ratio estimated from EDX graph.**

We thank the reviewer for the suggestion. We have included all the EDX spectra in the manuscript. A succinct table and explanation have been given correlating the experimental stoichiometry of the synthesized compounds with the expected stoichiometry.

→ **It not possible to accept the VSM and Magnetization versus temperature graph**

We thank the reviewer for this suggestion. We have subtracted the blank to minimize the appearance of the instrument's noise. Also a smooth fit has been drawn for the magnetization versus temperature graph. The Barkhausen effect makes the curves not to be absolutely smooth.

→ **It is difficult to measure the Saturation Magnetisation by this VSM. Experiment should be performed with high magnetic field like 75 Koe. Temperature dependent Magnetization should be done for all the samples with and without Field cooling.**

We thank the reviewer for the comment. We have used the approach to saturation at high fields by plotting M vs $1/H$ and extrapolating to $1/H$ equals zero to obtain the saturation magnetization, M_s . In the work, temperature dependence on magnetization was carried out on the undoped compound, NdFeO_3 to illustrate the behavior of the ordered system. For the doped samples, we have noted the suggestion and will take it into account in perspective work.

Once again, we appreciate the reviewer's comments and suggestions. The results obtained have been explained in greater details with the appropriate references.

Reviewer: 2

Comments to the Author(s)

The manuscript submitted by Nforna et al. report the "Effect of B-site Co substitution on the structure and magnetic properties of nanocrystalline neodymium orthoferrite synthesized by auto-combustion." In this report, authors have synthesized cobalt-doped neodymium orthoferrite compounds, $\text{NdCo}_x\text{Fe}_{1-x}\text{O}_3$ ($0.0 \leq x \leq 0.5$) using auto-combustion method which is different from the method have been reported previously to prepare $\text{NdCo}_x\text{Fe}_{1-x}\text{O}_3$. The powder XRD, EDX and magnetic

measurements were performed which shows difference in structural and magnetic behavior of the reported compounds in comparison to the undoped NdFeO. However, effect of doped Co amount on the both (structural and magnetic) follows an ambiguous trend and not well understood. Additionally, similar type of works has been previously reported. The manuscript needs several changes before publication in the high standard Royal Society Open Science Journal.

No IR Figure have shown in main text and SI, author should put IR figure of all reported samples in SI.

We thank the reviewer for the suggestion. We have included the FTIR spectra for all the samples in the SI.

In Figure 1 caption, author should write the formula correctly.

The formula has been written correctly.

It is obvious that the difference in stretching vibration of M–O in NCF19 and NCF55 is about 30 cm⁻¹. Can the authors explain the reason?

We thank the reviewer for the comment. We have explained the reason in the main text, as arising from shorter and stronger bonds as correlated with the XRD results.

The Fe–O–Fe bond angles for all reported compounds lie in the range of 150 -160° (except NCF46) indicating significant deviation in FeO₆ octahedral geometry not slightly. Author should modify the sentence accordingly.

We thank the reviewer for this suggestion. The authors after noticing this decided to further refined all the XRD patterns and obtained better refinement results.

Kindly change the format of Figure 3, 3a and 3b for NFO and NCF55 according to the text which is just reverse in Figure 3.

We have given the correct numbering.

It is obvious from the PXRD data and magnet data that there are several mismatches in the reported trend in structural and magnetic properties (e.g. particle size, M–O distances, Fe–O–Fe angles, MS, MR, Hc, etc.) upon Co doping. Can the authors explain the reasons?

We thank the reviewer for the question. We have reviewed most of the analyses. The trends have been explained in the text with any anomaly explained and appropriate references have been mentioned.

Why the Hysteresis loops of NdCo_xFe_{1-x}O₃ at room temperature (Figure 4) are so noisy?

We have deducted the signals of a blank from the measured samples in order to minimize the instrument's noise. We however, think that the Barkhausen effect is in play and is strongly affected by the microstructure and strain of the samples.

Figure 5. Magnetization versus temperature graph of NFO to determine the Neel temperature, they need to remeasure this data. This data is not acceptable in this current form. From this data it is impossible to find out the Neel temperature at 421 °C (694 K).

We thank the reviewer for the suggestion. As said above, we have tried to minimize the instrument's noise. A smooth fit for the magnetization vs temperature curve has been obtained which clearly outlines the shape and features of the graph.

Co³⁺ in LS state is diamagnetic and have no contribution to the magnetic properties. Then how it affects the exchange interactions?

We thank the reviewer for the question. We have explained in greater details in the text how the magnetic properties change and reason for the trend with references. In summary, the microstructure, particle size, pores and grain boundaries are mainly responsible for the variations observed in magnetic properties.

There are several errors in reference number, Table number, and Figure number formats.

We thank the reviewer for the remark. We have corrected the errors in the reference, table and figure numberings.

In general, many spaces are missing in between the words in the text. Kindly check this.

We appreciate the reviewer for this general observation. We have done our best to address this.

We extend our sincere gratitude to the reviewers and the editorial board members for the positive evaluation of the work for improvement. We have addressed all the points mentioned to our possible best and hope the ameliorated manuscript is good enough in the present format and merits publication.

Thanks,
Yours sincerely

The authors.